

# Satellite remote sensing of environmental variables can predict acoustic activity of an orthopteran assemblage

Diego A. Gomez-Morales[1,2] and Orlando Acevedo-Charry[3,4]

[1] Departamento de Biología, Universidad Nacional de Colombia, Bogotá, Bogotá D.C., Colombia
[2] Department of Biology, California State University, Northridge, California, United States
[3] Colección de Sonidos Ambientales *Mauricio Álvarez-Rebolledo*, Colecciones Biológicas, Subdirección de Investigaciones, Instituto de Investigación de Recursos Biológicos Alexander von Humboldt, Villa de Leyva, Boyacá, Colombia
[4] School of Natural Resources and Environment, Department of Wildlife Ecology and Conservation & Florida Museum of Natural History, University of Florida, Gainesville, Florida, United States

Corresponding author
Diego A. Gomez-Morales,
diego.gomez.519@my.csun.edu

## ABSTRACT

Passive acoustic monitoring (PAM) is a promising method for biodiversity assessment, which allows for longer and less intrusive sampling when compared to traditional methods (*e.g.*, collecting specimens), by using sound recordings as the primary data source. Insects have great potential as models for the study and monitoring of acoustic assemblages due to their sensitivity to environmental changes. Nevertheless, ecoacoustic studies focused on insects are still scarce when compared to more charismatic groups. Insects' acoustic activity patterns respond to environmental factors, like temperature, moonlight, and precipitation, but community acoustic perspectives have been barely explored. Here, we provide an example of the usefulness of PAM to track temporal patterns of acoustic activity for a nocturnal assemblage of insects (Orthoptera). We integrate satellite remote sensing and astronomically measured environmental factors at a local scale in an Andean Forest of Colombia and evaluate the acoustic response of orthopterans through automated model detections of their songs for nine weeks (March and April of 2020). We describe the acoustic frequency range and diel period for the calling song of each representative species. Three species overlapped in frequency and diel acoustics but inhabit different strata: canopy, understory, and ground surface level. Based on the acoustic frequency and activity, we identified three trends: (i) both sampled cricket species call at lower frequency for shorter periods of time (dusk); (ii) all sampled katydid species call at higher frequency for longer time periods, including later hours at night; and (iii) the diel acoustic activity span window seems to increase proportionally with dominant acoustic frequency, but further research is required. We also identified a dusk chorus in which all the species sing at the same time. To quantify the acoustic response to environmental factors, we calculated a beta regression with the singing activity as a response variable and moon phase, surface temperature and daily precipitation as explanatory variables. The response to the moon phase was significant for the katydids but not for the crickets, possibly due to differences in diel activity periods. Crickets are active during dusk, thus the effects of moonlight on acoustic activity are negligible. The response to precipitation was significant for the two crickets and not for the katydids, possibly because of higher

likelihood of rain interrupting crickets' shorter diel activity period. Our study shows how the local survey of orthopteran acoustic assemblages, with a species taxonomic resolution coupled with remote-sensing environmental measurements can reveal responses to environmental factors. In addition, we demonstrate how satellite data might prove to be a useful alternative source of environmental data for community studies with geographical, financial, or other constraints.

## INTRODUCTION

Many animals use acoustic signaling as their principal form of communication (*Bradbury & Vehrencamp, 2011*), contributing to the biotic component of a soundscape (*Pijanowski et al., 2011*). Ecological questions regarding behavior, seasonal activity, or response to external factors at different ecological and temporal scales of acoustic communities can be addressed with acoustic monitoring (*Sugai et al., 2019*; *Gottesman et al., 2021*; *Chhaya et al., 2021*); such monitoring can be with in-person recording in the field (*Drewry & Rand, 1983*; *Diwakar & Balakrishnan, 2007a*) or with automatic, passive recording (*Deichmann et al., 2018*). Favored by the increasing access to new recording technologies and computational power, passive acoustic monitoring (PAM) has become one of the trending methods to obtain environmental recordings (*Riede, 2018*; *Sugai et al., 2019*). PAM consists of the deployment of autonomous passive recorders in the field. Its advantages for biodiversity monitoring including longer term assessment periods, less intrusive monitoring methods, increase of data collection, and increased potential for community bioacoustics research at different scales, when compared to classical monitoring approaches such as specimen collection in the field (*Blumstein et al., 2011*; *Deichmann et al., 2018*; *Sugai et al., 2020*). Furthermore, PAM allows the classification of calling songs into recognizable taxonomic units, also referred to as acoustic morphospecies or sonotypes (*Riede, 1998*; *Aide et al., 2013*; *Ferreira et al., 2018*). Despite that, there are still challenges when applied to high diverse taxonomic groups with lees availability of taxonomic and acoustic descriptions, as insects (*Riede, 2018*). To address these challenges, we acoustically monitored a nocturnal orthopteran assemblage and matched the sonotypes with taxonomic species identified from voucher specimens captured at the same location.

Insect sounds drive tropical soundscapes by contributing acoustic signaling that varies in time, acoustic frequency, and spatial scales (*Aide et al., 2017*). In addition, insect sounds have high potential as acoustic bioindicators, especially at local scales, due to their sensitivity to environmental change (*McGeoch, 2007*; *Jeliazkov et al., 2016*; *Riede, 2018*). Within insects, crickets and katydids (Orthoptera: Ensifera) are some of the most important acoustic contributors to soundscapes. They produce sound by rubbing together specialized structures in their wings, a behavior called elytral stridulation (*Baker & Chesmore, 2020*). Between the different types of stridulatory calls, the "calling song" is used by males to attract the opposite sex (*Grimaldi & Engel, 2005*; *Buellesbach, Cash & Schmitt,*

*2018*). Calling songs are the most common insect signals to be found in natural soundscapes, and they are often used in taxonomic and evolutionary studies (*Tan et al., 2021*) due to their stereotyped characteristics and high species-specificity (*Grimaldi & Engel, 2005*). However, few ecoacoustics studies include detailed taxonomic resolution for orthopterans or other insects (*Diwakar & Balakrishnan, 2007a*; *Gasc et al., 2018*), and many community approaches are rather focused on charismatic vertebrates such as birds (*Tobias et al., 2014*), anurans (*Villanueva-Rivera, 2014*), or mammals (*Heinicke et al., 2015*). When it comes to the community scale, matching acoustic species with taxonomic species will help to unveil biological communities' structure and change over space and time (*Chhaya et al., 2021*).

At a community level, the changes in acoustic activity patterns can be explained by the influence of a set of biotic and abiotic factors (*Chhaya et al., 2021*). On one hand, explanations of how biotic factors influence acoustic patterns have been formalized by different theories (*Farina, 2014*). The acoustic niche hypothesis for example, states that animals respond to other signalers by partitioning their acoustic activity (*Krause, 1993*) in order to avoid acoustic masking (or overlap) in time, spectral frequency, and space, thus optimizing the signal transmission (*Schmidt, Römer & Riede, 2013*). However, high time overlap is usual in tropical orthopteran assemblages, especially during dusk chorus (*Riede, 1996*; *Jain et al., 2014*). The tendency of animals from different species to concentrate their singing at the same time window has been previously referred to as clustering (*Tobias et al., 2014*), and may benefit individual singers by lowering predation risks (*Jain et al., 2014*; *Farina & Ceraulo, 2017*). On the other hand, abiotic environmental factors such as temperature, precipitation or moonlight are known to predict acoustic patterns in vertebrate animal communities (*Bruni, Mennill & Foote, 2014*; *Pérez-Granados, Schuchmann & Marques, 2022*), as well as insects (*Lang et al., 2006*; *Franklin et al., 2009*). Acoustic activity can be a very useful description of circadian and seasonal patterns (*Aide et al., 2013*), and species responses to environmental variables (*Pérez-Granados, Schuchmann & Marques, 2022*).

Most studies at community ecological scales gather environmental information locally, by directly measuring the variables in the field, or by gathering it from local weather stations (*Ospina et al., 2013*). However, field environmental information could be difficult or even impossible to obtain in some circumstances, due to the sparse distribution or absence of weather stations in some areas, remote location of study sites, or budget constraints. Satellite remote sensing could be an alternative in such cases because it is spatially and temporally comprehensive, despite lower resolution. Data gathered from satellite remote sensing has been useful for ecological studies at landscape or habitat levels (*Pettorelli et al., 2014*; *Pasetto et al., 2018*), but its potential for local community scale questions, such as the relationship with acoustic activity, remains unexplored.

Here, we characterize the acoustic activity of an orthopteran assemblage, measuring occurrence of calling events per time unit and the level of inter-specific overlap in temporal and frequency range. We expected to find a cluster of singing species during dusk (dusk chorus), but no species overlapping in all three dimensions (time, frequency, and space). We also decided to explore the implementation of satellite remote sensing data of

PeerJ ________________________________________

environmental variables (surface temperature, daily precipitation) along with astronomical calculated moon illuminated fraction, to describe their species-specific effect on acoustic activity of an orthopteran assemblage in the tropical Andes of Colombia. In addition, we recorded and collected insects (including some new species for science that are being described) to identify sonotypes to species taxonomic level (when possible). This work serves as an important basis for future acoustic monitoring protocols of insects by identifying acoustic bioindicators and using available remote sensing data to provide further insight for localized community ecology studies.

## MATERIALS AND METHODS

### Study site and acoustic sampling

We sampled the soundscape at the Los Tucanes Natural Reserve, located in Gachantivá, Boyacá, Eastern Andes of Colombia (5.789, −73.550; 2300 ± 25 m asl). This private reserve includes an area of 0.16 km² of sub-Andean Forest dominated by Andean oak (*Quercus humboldtii*) in different regeneration states, with an annual temperature of 15 °C. Annual precipitation is about 1,450 mm, in two rainy seasons: from March to May and from October to November (*Climate-Data.org, 2019*). Overall, this forest is a ~20 years old secondary forest with early successional grasslands after agricultural abandonment. We used the acoustic monitoring data from an Audiomoth 1.0.0 recorder deployed at a height of 1.5 m from the ground, with a sampling rate of 48 kHz and resolution of 16 bits, set to record for 1 min every 30 min during the rainy season, from March to May of 2020 (*Tovar Garcia & Acevedo-Charry, 2021*).

    The recording set (number of recordings, $n = 2,851$) was normalized to −3 dB and uploaded to the ARBIMON online platform by Rainforest Connection-RFCx (*Arbimon, 2020*), download information available in Data S11. From there, we manually explored a subsample of recordings (Table S1, training set size) to annotate the presence or absence of the acoustic species most consistently observed throughout the days. We defined sonotypes before identifying taxonomic species (see below) which were then used during the data analysis. Other sonotypes with sporadic acoustic activity were not considered in the analyses given the difficulty of training detection models from very few annotated recordings. Then, we trained random forests (Datas S1, S2, S4, S6) and spectrogram template matching (or pattern matching) (Datas S3, S5, S7) automatic recognition models using RFCx ARBIMON integrated tools for each sonotype, using the annotated recordings as the training dataset (Table S1) in order to detect the occurrences of sonotypes for every recording. We determined that the detection performance of spectrogram pattern matching was better for katydids, while random forests was significantly better for crickets based on preliminary tests. A detailed performance comparison between available detection models (including those made for other animal groups) for orthopteran species remains as an interesting issue to address, the more when new promising tools have recently been launched (*e.g.*, *Lapp et al., 2021*; *Steinfath et al., 2021*). Model output was manually revised, false positives were discarded, and detection precision was calculated as suggested by *Aide et al. (2013)*: all detection models include a precision above 70% (Table S2). After post validation, the model output was a presence/absence per recording

matrix for the complete recording dataset. We defined the acoustic activity as the presences detected per time unit: hours and days (*Aide et al., 2013*). Using the acoustic activity per hour, we described the average daily activity per species and measured the temporal partition between pairs of species by the overlap of kernel densities, which is a coefficient that reflects extent of overlap between activity patterns, as a measure of similarity (*Ridout & Linkie, 2009*). In addition, we use the activity per day to fit models of satellite remote-sensed environmental variables (see below).

To assign sonotypes with better taxonomic resolution, we collected specimens in the field and conducted taxonomic delimitation by actively looking for the emitters of each signal of sonotypes selected (those most consistently observed in PAM dataset) during two field trips, from September and October 2020. Observations of microhabitat (height) and singing behavior were made during these field trips. The collected specimens were deposited at the Instituto Humboldt's entomological collection (IAvH-E) in Villa de Leyva (Boyacá) following extended specimens' guidelines (*Acevedo-Charry et al., 2021*), with a genetic voucher (foreleg) also deposited at the Instituto Humboldt's tissue collection in Palmira (Valle del Cauca, Colombia). Specimens were identified to the highest taxonomic resolution possible. For some species, we were unable to collect specimens (*i.e.*, canopy dwellers), thus we assigned a sonotype temporary name (*e.g.*, "Flutist") to the acoustic signal (*Aide et al., 2017*; *Ferreira et al., 2018*). In addition, we made recordings from some individuals in captivity (species: Katydid1, Katydid3, Katydid4) or directly in the field (Katydid5) with an Audiomoth 1.0.0 recorder at a sampling rate of 384 kHz and a resolution of 16 bits. Reference recordings for other species (Cricket1, Cricket2 and Katydid2) were taken in the field using a Sennheiser ME67 shotgun microphone attached to a first generation SoundDevice Mix Pre 3 recorder. Acoustic recordings were deposited in the Instituto Humboldt's sound collection-Colección de Sonidos Ambientales *Mauricio Álvarez-Rebolledo* (IAvH-CSA-18783 to IAvH-CSA-18805).

## Environmental variables from remote sensing data

With the detection dates across our study time, we evaluated the relationship of the acoustic activity per day of each species with satellite-detected local temperature, precipitation, and moonlight. We extracted the time series from the pixel overlaying the sampling point from daily generated raster files using the software Quantum GIS (*QGIS Development Team, 2022*). Temperature was obtained from a 1 km resolution dataset using the land surface temperature (LST) parameter and generated using moderate resolution imaging spectroradiometer (MODIS) LST products (*Zhang et al., 2022*); this dataset includes a measurement at 01:00 and another at 13:00 h, equivalent to minimal and maximal daily temperature. On the other hand, precipitation values were obtained from a 10 km resolution dataset estimated using the Integrated Multi-satellitE Retrievals for Global Precipitation Measurement (IMERG) (*Huffman et al., 2019*). Environmental data used for the analysis is provided in Data S8. Preliminary analysis showed high correlation with the nearest national station at Santa Sofía, Boyacá (10 km away of our study site) but IMERG data include more steady values (*i.e.*, no gaps in days as Santa Sofía had). Finally,

moon illuminated fraction was retrieved through the function *getMoonIllumination* from the R package *suncalc* for our sampling site coordinates (*Thieurmel & Elmarhraoui, 2019*).

## Data analysis

For temporal variables, diel acoustic activity of each species was analyzed using the package *overlap* in R (*Ridout & Linkie, 2009*). First, we generated the von Mises kernel density distribution of the diel acoustic activity for each species during the complete sampling period (March to April) given that daily acoustic activity patterns correspond to a circular distribution. Then, we computed the delta coefficient ($\hat{\Delta}_4$) of overlapping between every pair of species, as recommended for samples sizes bigger than 75 (*Ridout & Linkie, 2009*) with the default smoothing value. Confidence intervals were calculated by using a smoothed bootstrap with 10,000 samples, adjusted for bias (*Ridout & Linkie, 2009*).

Spectral variables, i.e., dominant frequency (frequency with the most energy), frequency bandwidth from minimum ($-5$ dB below dominant frequency) to maximum frequencies ($+5$ dB over dominant frequency), were measured manually from the FFT analysis window (Hamming window, 256 samples) of the software *ocenaudio* (*Ocenaudio Team, 2015*) for the dominant harmonic of 18 syllables, as defined by *Baker & Chesmore (2020)* for katydid species, and eighteen 1s segments for cricket species. Segments were selected from the recordings with less noise from the passive recording dataset for each species (Data S10). Additional harmonics were not measured as they were not constantly detected by the recorders due to attenuation (*Romer & Lewald, 1992*; *Hung & Prestwich, 2004*). To assess frequency overlap, we contrasted the measurements (minimum, maximum and dominant frequencies) for each species by randomly resampling the measurements with 10,000 iterations. Each iteration per species and measurement includes a mean value and confidence intervals of 95% of the data (Table S3).

We explored the relationship of the proportion of detections of each species in each day with the four continuous remote-sensed variables through a beta regression, using the package *betareg* in R (*Cribari-Neto & Zeileis, 2010*). We assumed a beta distribution because the values were continuous and bounded between 0 and 1 (*Bolker, 2007*), scaling the occurrence detections with the formula $Detections_{(b)} = [Detections_{(a)} * (N-1) + 0.5]/N$, where $N$ is the sample size and $Detections_{(b)}$ is our response variable for the beta regression (*Smithson & Verkuilen, 2006*). To select the variable that better fits the response variable (proportion of detections) for each species, we scaled the predictor variables (min and max temperature, precipitation, and moon fraction) by centering around the mean and dividing by two standard deviations (*Schielzeth, 2010*). For each species, we compared 16 additive models with different combinations of predictor variables, including a null model. We identified the most frequency top-ranked model ($\pi_i$) based on the delta Akaike Information Criteria ($\Delta$AIC) and Akaike weight ($w_i$) after resampling 10,000 times. To estimate better the top ranked model, in each resampling we randomly select only 50 of the 61 days of recordings and estimate the percentage of times each top-ranked model was selected ($\pi_i$) (Data S9). We conducted our analyses in R, and our code and data are available (https://github.com/diegryllid).

## RESULTS

We focused on seven identified orthopteran species for our analysis of the acoustic assemblage. Two species of crickets (superfamily Grylloidea): "Flutist" (Cricket1) and Podoscirtinae (Cricket2) and five katydids (family Tettigoniidae): *Copiphora colombiae* (Katydid1), *Neoconocephalus brachypterus* (Katydid3), Cocconotini (gen. nov.) (Katydid4), "Sprinkler" (Katydid2), and "Rattler" (Katydid5). Although not all the species in the acoustic assemblage were covered, these seven species were the most representative ones in terms of acoustic activity (the ones detected most of the days) along the sampling period, as defined after the manual annotation of recordings.

### Male singing behavior, microhabitat partitioning and songs of the assemblage

We detected species dwelling in different microhabitats by sonotype *in situ*. In the upper side of the forest, we detected the canopy dwellers (Cricket1, Katydid2, Katydid5). In the middle strata, among shrubs and low tree branches (<4 m), we detected Podoscirtinae (Cricket2). Similarly, we detected Cocconotini gen. nov. (Katydid4) mostly at heights of ~4 m at most. Finally, we detected Katydid1 and Katydid3 in more open areas and early successional shrubs. *Cophiphora colombiae* (Katydid1) sang from underneath the leaves or on top of the stems of understory shrubs around 2 m above the ground, while *N. brachypterus* (Katydid3) always sang from clumps of grass, very close to the ground (0.5–1 m). The heights here referred are visual estimates, hence we recommend direct measurement of heights of individuals for a detailed description of spatial distribution in forest strata. We detected two main singing behaviors as well. The first one was a cryptic singing behavior consisting of singing undercover, observed in two species (Cricket2 and Katydid4). The cricket species Podoscitinae (Cricket2) always sang from inside rolled oak leaves, while the katydid species Cocconotini gen. nov. (Katydid4) sang from its own carved burrow in live tree trunks (O. Cadena-Castañeda, D.A. Gòmez-Morales, O. Acevedo-Charry, J.L. Benavides-Lòpez, 2022, Unpublished manuscript). The second behavior was singing exposed, consisting of calling from the top of leaves and branches, observed in two katydid species: *C. colombiae* (Katydid1) and *N. brachypterus* (Katydid3). We could not observe singing behavior for the canopy dwellers (Cricket1, Katydid2, Katydid5).

### Diel acoustic activity

Diel acoustic activity varied across the night, reflecting two strategies. All katydids have a completely nocturnal singing behavior (Katydid1, Katydid2, Katydid3, Katydid4, Katydid5), although "Rattler" (Katydid5) showed a few, sparse detections during the day (Data S7). These five species started singing at dusk (~17:30 h); then, three species showed a higher density of detections in specific times of the night (Katydid1: ~20:30, Katydid2: ~01:30, Katydid3: ~20:00) shown as the peak acoustic activity density (Fig. 1). It is important to point out that these three species are still active throughout the whole night, and their activity peaks are subtle (less than 0.1 difference in acoustic activity density when compared to the mean of the rest of the hours of activity). The remaining two katydids

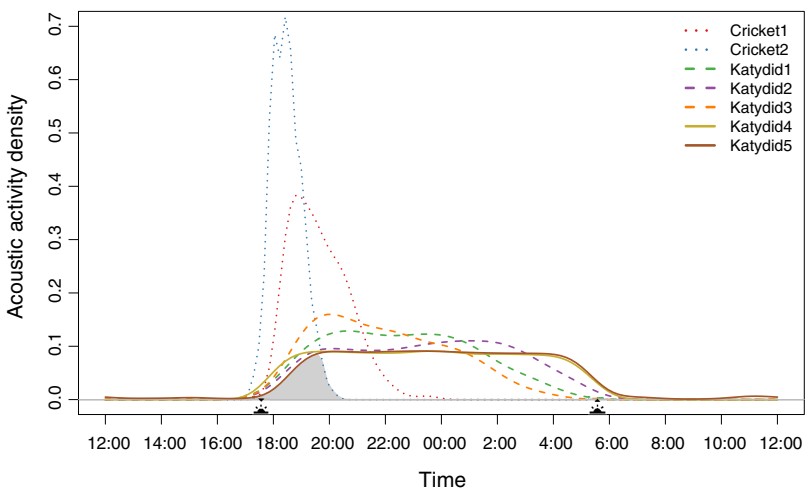

**Figure 1 Acoustic activity density per species.** The distribution was calculated from the number of detections every half an hour during the 2-month sampling period. The acoustic activity density value corresponds to the proportion of positive detections at a given time. The shaded area demarks the probability of a dusk chorus when all species sing at the same time. The sun icons represent the sunset time and sunrise times. The type of line denotes the dominant frequency range: solid for high frequency, dashed for medium, and dotted for low.

**Table 1 Kernel density delta coefficient for pairs of species. Bootstrap 95% confidence intervals in parenthesis. The values go from 0 to 1, 1 indicating complete overlap.**

|  | *Podoscirtinae* (Cricket2) | *C. colombiae* (Katydid1) | "sprinkler" (Katydid2) | *N. brachypterus* (Katydid3) | Cocconotini (gen. nov); (Katydid4) | "Rattler" (Katydid5) |
|---|---|---|---|---|---|---|
| **"Flutist" (Cricket1)** | 0.49 (0.41–0.58) | 0.42 (0.36–0.47) | 0.34 (0.30–0.38) | 0.51 (0.44–0.57) | 0.36 (0.32–0.40) | 0.32 (0.28–0.36) |
| ***Podoscirtinae* (Cricket2)** |  | 0.16 (0.12–0.20) | 0.14 (0.11–0.18) | 0.21 (0.16–0.25) | 0.17 (0.14–0.20) | 0.12 (0.09–0.15) |
| ***C. colombiae* (Katydid1)** |  |  | 0.86 (0.81–0.90) | 0.89 (0.83–0.94) | 0.81 (0.76–0.85) | 0.78 (0.74–0.82) |
| **"sprinkler" (Katydid2)** |  |  |  | 0.77 (0.71–0.81) | 0.91 (0.88–0.94) | 0.90 (0.87–0.93) |
| ***N. brachypterus* (Katydid3)** |  |  |  |  | 0.74 (0.70–0.78) | 0.72 (0.67–0.76) |
| **Cocconotini (gen. nov); (Katydid4)** |  |  |  |  |  | 0.95 (0.92–0.97) |

(Katydid4, Katydid5) showed relatively constant acoustic activity throughout the night and a faster decrease in activity from 04:00 until sunrise (~06:00). On the other hand, both crickets have a crepuscular schedule, singing mainly during dusk. Podoscirtinae (Cricket2) showed a much more restricted singing schedule, from 18:00 to 19:00 with a very abrupt decrease in activity afterwards. Flutist (Cricket1) showed a slower decrease in activity, maintaining a considerable number of detections after ~21:00. Overall, we identified a chorus time when all the members of the assemblage sing at the same time, from 18:00 to 20:00 (Fig. 1).

## Diel and spectral overlap

Species density distribution pairwise analysis showed two overlap trends (Table 1). Crickets have less probability of overlapping with the katydids' singing schedules (coefficient values <0.6). Of these two, Podoscirtinae (Cricket2) was the most specialized,
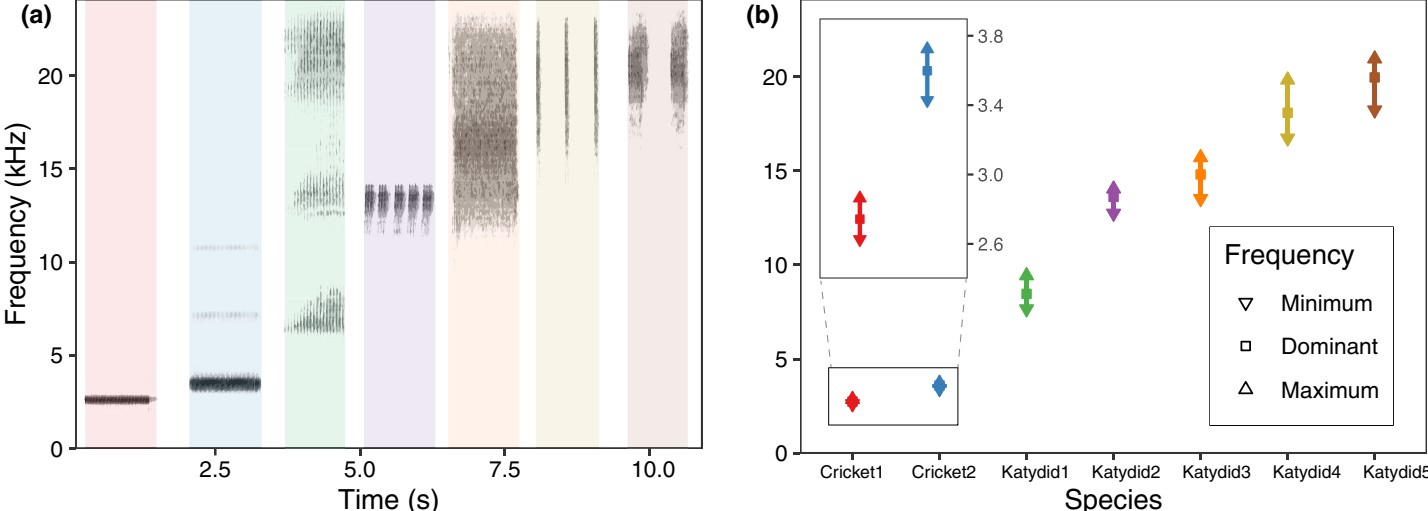

**Figure 2 Frequency range per species.** (A) Spectrogram sections for the study species at 48 kHz, same species order as in (B) average values for minimum, maximum, and dominant frequency calculated after a bootstrap sampling of 10,000 iterations.

having a very short singing schedule (Fig. 1). Conversely, katydids overlap in their singing periods (coefficient values >0.6).

The mean dominant frequency was considerably lower for crickets, and higher for katydids. While cricket bandwidth was narrow, katydid frequency bandwidth varied across species, reflecting spectral overlap in four species (Fig. 2). The species which used a higher frequency also include a higher bandwidth (Katydid5, Katydid4, Katydid3), overlapping among them. The species with higher bandwidth, *N. brachypterus* (Katydid3), overlapped additionally with a canopy dweller species, Katydid2, which included a lower frequency bandwidth. Most of the species had harmonic components in their calling song (all excepting Cricket1) as observed in the directional recordings (Fig. 2A) with ultrasonic harmonics for all the katydids except Katydid2. (*Romer & Lewald, 1992*; *Hung & Prestwich, 2004*). Ultrasonic harmonics were not recorded or analyzed because of the sampling rate used in our PAM setting; audible harmonics other than the dominant one (the harmonic with the highest amplitude) where not analyzed either, as their lower amplitude made them undetectable in the automated field recordings.

## Acoustic activity response to environmental variables

The response of the acoustic activity varied by species and environmental variables obtained from satellite remote sensing sources (Fig. 3, Table 2). Acoustic activity of katydid species included a negative relationship with moon fraction for Katydid1 ($\beta = -0.28$, $p = 0.003$), Katydid2 ($\beta = -0.34$, $p < 0.001$), Katydid3 ($\beta = -0.36$, $p < 0.001$), Katydid4 ($\beta = -0.13$, $p < 0.001$), and Katydid5 ($\beta = -0.07$, $p = 0.02$). Both cricket species showed a negative relationship only with the precipitation (Cricket1: $\beta = -0.31$, $p = 0.006$; Cricket2: $\beta = -0.32$, $p = 0.002$) while in katydids the response was more complex. In addition to the cricket species, precipitation was also strongly negatively related to acoustic activity of Katydid1 ($\beta = -0.25$, $p = 0.02$), but less for Katydid5 ($\beta = -0.06$, $p = 0.05$). Minimum
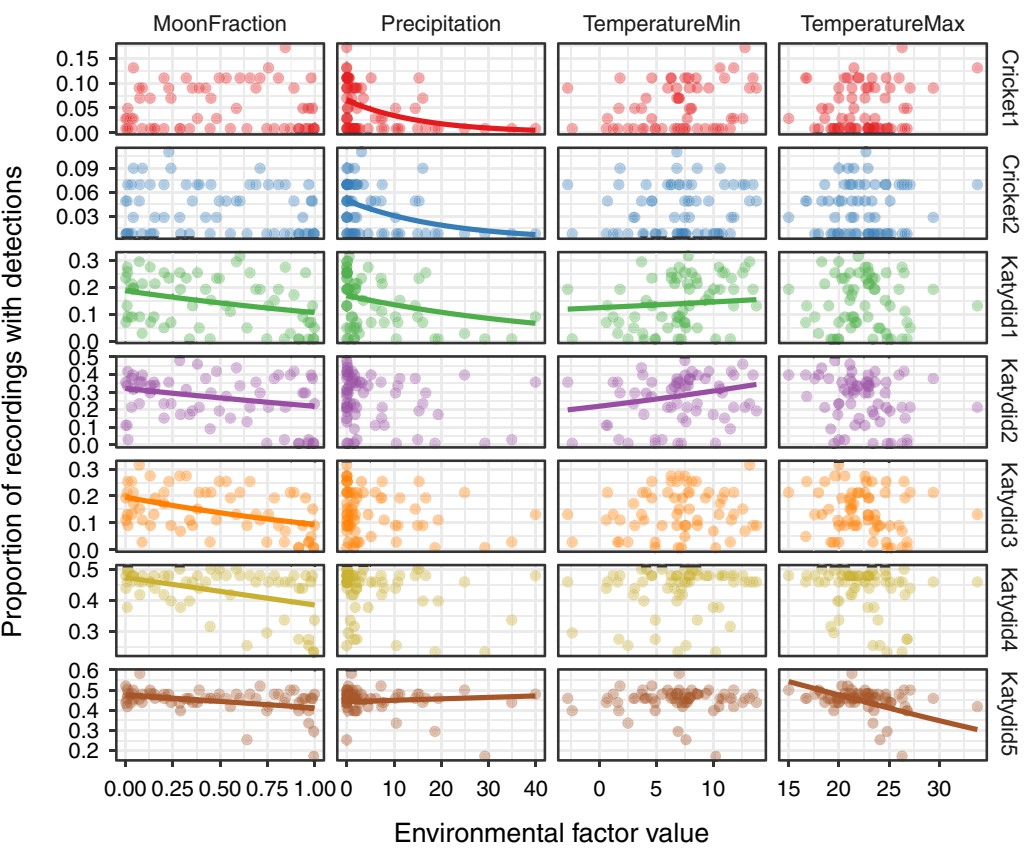

**Figure 3 Relationship of acoustic detections with environmental factors.** For each of the seven focal species listed on the right axis, the inset figures show the proportion of acoustic detections as a function of four environmental variables measured *via* remote sensing: MoonFraction, Precipitation, Minimum Temperature, and Maximum Temperature. The points represent the proportion of positive detections per day during the 2-month sampling period. Trend lines are shown for the significant relationships as determined by the beta regression analysis.

**Table 2 Results of beta regressions for the seven species.**

| Species | Model (β) | $\pi_i$ | $k$ | $w_i$ | LL | Φ | z-value | Pr (>\|z\|) |
|---|---|---|---|---|---|---|---|---|
| Cricket1–"flutist" | Precipitation (−0.31) | 76.2 | 3 | 0.28 | 125.80 | 22.89 | 5.09 | <0.001 |
| Cricket2–Podoscirtini | Precipitation (−0.32) | 94.8 | 3 | 0.35 | 142.70 | 41.49 | 5.22 | <0.001 |
| Katydid1–*C. colombiae* | MinTemp (0.26) + Precipitation (−0.25) + Moon (−0.28) | 82.4 | 5 | 0.42 | 73.64 | 13.37 | 5.43 | <0.001 |
| Katydid2–"sprinkler" | MinTemp (0.21) + Moon (−0.34) | 30.2 | 4 | 0.25 | 38.30 | 7.00 | 5.69 | <0.001 |
| Katydid3–*N. brachypterus* | Moon (−0.36) | 56.8 | 3 | 0.28 | 73.63 | 16.2 | 5.49 | <0.001 |
| Katydid4–Cocconotini (gen. nov.) | Moon (−0.13) | 61.7 | 3 | 0.26 | 77.04 | 51.01 | 5.58 | <0.001 |
| Katydid5–"rattler" | MaxTemp (−0.07) + Precipitation (−0.06) + Moon (−0.07) | 62.3 | 5 | 0.27 | 89.25 | 77.43 | 5.56 | <0.001 |

**Note:**
Response variable is the proportion of detections per day. Explanatory variable models include minimal (MinTemp) and maximal temperature (MaxTemp), daily precipitation, and moon fraction (Moon), in parenthesis is shown the estimated slope for the log odds ratio of each variable (β). $\pi_i$ shows the percentage of times a model $i$ was top-ranked after 10,000 iterations. $k$ is the number of estimated parameters. $w_i$ is the Akaike weight. Φ, SE (standard error), LL (log-likelihood), z-value, and Pr (>\|z\|) shows the result of the beta regression for the top-ranked model (see beta regression results in Data S8).

temperature showed a positive trend for both Katydid1 (β = 0.26, $p$ = 0.007) and Katydid2 (β = 0.21, $p$ = 0.04), and maximum temperature was significantly negatively related for Katydid5 only (β = −0.07, $p$ = 0.03). In general terms, daily precipitation is a better

explanatory for the acoustic activity of crickets, while moon fraction for acoustic activity of katydids.

## DISCUSSION

Despite insects' acoustic activity being driving the soundscapes of tropical ecosystems (*Aide et al., 2017*), they are rarely identified with accurate taxonomic resolution in ecoacoustic studies. We used the acoustic signaling footprint of different species from passive acoustic sampling to characterize an orthopteran assemblage. With the daily acoustic activity pattern for each sonotype, we were able to sample the prioritized species and match four out of seven species. Despite recent improvement in taxonomic work in the group for Colombia (*Cadena-Castañeda et al., 2020*, *2021*), and impressive research on sound production and reception (*Baker et al., 2019*; *Celiker, Jonsson & Montealegre-Z, 2020*), we found taxonomic novelties from this community (O. Cadena-Castañeda, D.A. Gòmez-Morales, O. Acevedo-Charry, J.L. Benavides-Lòpez, 2022, Unpublished manuscript). In addition, we explored the relationship of acoustic activity of each species with environmental variables extracted from satellite remote sensing data. We experienced difficulties in collecting canopy dwellers, for which we recommend the use of specialized methods such as fogging (*Montealegre-Z et al., 2014*), light trapping (*e.g.*, *Symes et al., 2021*), or specialized manual tracking (*Diwakar & Balakrishnan, 2007b*) for future studies. As studies on environmental effects on insect acoustic communities are still rare, our study is an important precedent, and serves as a good example on how satellite remote sensing data can be used along with acoustic monitoring schemes in areas with low accessibility to ground-based methods of environmental measurement, such as weather stations.

### Audible representative species

Our focused species represent the most common nocturnal audible orthopteran insects in our sampling site. Our approach, focusing in low frequency (audible spectrum) conspicuous species has benefits and challenges. By focusing on the most common species: we were able to describe the assemblage sonotypes identity as well as the overall trend of singing behaviors that dominate the community. However, it left aside other species which are rare or undetectable by our methods. Diurnal species, for instance, had relatively low detectability within our data set (grasshoppers mostly) and were barely detected, which confirms previous observations for the group in tropical forests (*Diwakar & Balakrishnan, 2007a*). For studies with the aim of extensively describing an acoustic community, or analyzing detailed interactions between species, we recommend using more extensive sampling in both space (more recording units) and time (longer monitoring periods) along with more intensive annotation of recordings; for example, by using ultrasonic sampling for propper detection of some Neotropical katydid species. The use of multiple recording units can also enhance the detectability of low amplitude song species, which may have been under detected in our study. Despite that, given their high activity and detectability, combined with our ability to match taxonomic resolution of most of the species, we consider the representative species here taken into consideration are adequate and sufficient for analyzing single species relations with environmental factors for ecoacoustic
monitoring purposes. Therefore, we recommend this approach of species prioritization for environmental studies with time or methodological constraints.

## Time and spectral overlap

We observed that the higher the dominant frequency of their calling song (Fig. 2), the broader the diel acoustic activity range seemed to be (Fig. 1; Fig. S1). However, such pattern must be studied in detail in further research, focusing on the underlying mechanism regarding interspecific competition, morphological constraints, and/or predator-prey relationships. While cricket species had low frequency calling, narrowly clustered during dusk (Fig. 1, dotted line), katydids dispersed their calling widely throughout the night. Furthermore, medium frequency calling katydids (Fig. 1, dashed lines; Katydid1, Katydid2, Katydid3) show clear activity peak times, and the high frequency ones (Fig. 1, solid lines; Katydid4, Katydid5) maintain constant activity levels throughout the night. Previous studies found a negative relationship between calling signal duration and daily signaling rate in Neotropical katydids as result of acoustic trade-offs (*Symes et al., 2021*), our data suggests that there might be a positive relationship between the dominant frequency and the daily acoustic activity as well. In addition, although we did not measure body size or specific size and properties of the producing sound structures for the species in our assemblage, crickets were overall smaller than katydid species, thus our observation might be contrary to the predictions of the morphological adaptation hypothesis (*Farina, 2014*), previously proven in birds (*Wallschläger, 1980*), mammals (*Fletcher, 2004*), and frogs (*Boeckle, Preininger & Hödl, 2009*). Such a hypothesis may not even apply to orthopterans, given that their way of producing sound (elytral stridulation) is completely different. For example, *Godthi, Balakrishnan & Pratap (2022)* found a relationship of the size and properties of the stridulatory apparatus with acoustic frequency, independent to body size. Further examination of our vouchered specimens at IAvH-E might be useful for testing morphological hypotheses. Finally, previous studies have shown that nocturnal predators, like bats, eavesdrop and select their prey based on certain signal properties including peak frequency, and vary among species present in a community (*Falk et al., 2015*); then in this case, lower peak frequency species in this assemblage could be under stronger predation pressures, and respond by concentrating their singing in shorter time periods to lower that risk (*Farina & Ceraulo, 2017*). Further experimental research on these topics for insects remains needed.

In addition to the aggregation in time, there is a clear preference for dusk times by crickets, that can be explained by the fact that it is the time of the day when diurnal predators are already becoming inactive plus nocturnal predators being still not at their peak (*Jain et al., 2014*). Also, this is the time where all the katydid species start to become active as well, so the simultaneous interspecific singing can lower individual risk by "confusing" the predator who now has many choices (*Farina & Ceraulo, 2017*).

We previously referred to this period as the "dusk chorus", and many other factors may be influencing this phenomenon. Moreover, the relatively high katydid temporal overlapping (Table 1; Fig. 1) suggests that katydids are also aggregating their acoustic activity in time, only in a different fashion by concentrating on their activity later and

extending it during the whole night. Maintaining a continuous sing-along but extending their signaling for several hours also comes with its downsides, which require further adaptations in response to visual predators, as we will discuss later. Previous works have described how katydid aggregation in space can lower the effectiveness of bat captures (*Prakash et al., 2021*). Although we did not test the relationship between acoustic frequency and duration of the diel calling activity, our preliminary analysis suggests that species with lower dominant frequencies tend to call for shorter periods (*e.g.*, Cricket1 *vs* Katydid5; Fig. S1). Further research including more taxa and sampling locations and taking into account the mechanisms behind the dusk chorus along with the acoustic interactions with predators are necessary to confirm the "frequency-acoustic activity relationship". In addition, we did not account for phylogenetic or evolutionary constraints when contrasting these two groups of insects with very different limitations on the frequency of their songs, leaving open such research topic to better understand the orthopteran acoustic assemblage.

Even though four species show spectral overlap (Fig. 2), those might have different microhabitat preferences: Katydid2 seems to dwell in the canopy, while Katydid3 was always found at ground level grass chunks. Although we noted microhabitat observations based on opportunistic sampling, our results provide an approach to try to understand spectral and temporal overlap for some species. The species Katydid4 and Katydid5, which overlap in both time and frequency, inhabit the understory and canopy, respectively, and may be avoiding masking because of differential attenuation at different heights (*Ellinger & Hödl, 2003*). However, other studies have found conflicting evidence (*Jain & Balakrishnan, 2012*). Further research is required to test this hypothesis, as the effect of forest stratification over signal interference in insects remains unclear (*Schmidt & Balakrishnan, 2015*). In addition, measurements of smaller scale time features could help to confirm temporal masking avoidance at smaller scales (*Symes et al., 2021*), as Katydid4 and Katydid5 pulse rates and syllable duration seem to differ greatly.

## Environmental factors effects and Remote sensing data

Katydid species decrease acoustic activity 0.70 to 0.93 times with a unit increase in moon fraction light (from new moon at 0.0 to full moon at 1.0). The negative relationship found between moon fraction and katydid species acoustic activity (Table 2, Fig. 3) corroborates the findings of previous studies. For example, response to more moonlight led to avoidance of visual predators, either by lowering their overall activity (*Lang et al., 2006*), or increasing the use of alternate communication channels, such as tremulation (*Römer, Lang & Hartbauer, 2010*), as part of katydids' repertoire of adaptations for predation by bats (*ter Hofstede et al., 2017*). Response to moonlight is probably the consequence of the above-mentioned continuous time aggregation as opposed to crickets', which in fact, did not show any relationship to moonlight (Table 2), perhaps because during the dusk there is still plenty of sunlight, so moonlight would not make any difference.

Regarding precipitation, one cricket and two katydid species decrease acoustic activity 0.73 to 0.94 times with a unit increase in rainfall (mm per day). Such negative effect mainly over the acoustic activity of the two cricket species (Cricket1 and Cricket2; Fig. 3) confirms

previous observations (*Alexander & Meral, 1967*) and our own during the field: whenever it rained during the chorus time, the crickets barely sang. Although rain noise may have affected detectability on rainy days, our manual revision of recordings and observations in the field give us confidence in the model detections reflecting acoustic activity even during rainy days. Still, we recommend using acoustic detection models accounted for rain detection itself as a way of measuring rain impacts over detection of orthopterans in future studies. The katydids did not show that unique relationship with daily precipitation, except for Katydid1 and Katydid5 which included other covariates. However, something to consider is that given the shorter time span of crickets calling time, they had a greater chance of being interrupted by rain, as has been observed in other species (*Alexander & Meral, 1967*; *Franklin et al., 2009*), while broader katydid calling spans may have improved their detectability even after short rainy periods during the night. Our opinion is that rain may affect orthopterans singing at a finer time scale in addition to the accumulated daily effects, as opposed to moonlight, given that daily precipitation can be either sparsely or densely distributed during the day. In addition, other external factors such as wind (*Velilla et al., 2020*), or ultrasonic background noise (*Römer & Holderied, 2020*) can affect signaling behavior of katydid species as well. A comparison of both locally measured precipitation and satellite remote sensing data with community acoustic activity may be useful to confirm this relationship.

Temperature analysis suggests a negative relation (decrease 0.93 times with a unit increase) with daily maximum temperature only for Katydid5, and positive relation with minimum temperature for Katydid1 (increasing 1.30 times with a unit increase) and Katydid2 (increasing 1.23 times with a unit increase in min temperature). The underlying reasons for this response are yet to be discovered, but the increase in signaling by Katydid1 and Katydid2 species are concordant with previous studies in other species (*Franklin et al., 2009*) which found increased signaling during warmer nights. As the microhabitat's preference can play a fundamental role on temperature regulation, we rather recommend the use of local scale measurements to further explore calling activity relationship with this variable.

Although previous studies have evaluated the accuracy of the satellite remote sensing products here used or similar with environmental applications (*e.g.*, *Palomino-Ángel, Anaya-Acevedo & Botero, 2019*), or overviewed its potential for ecosystem modelling (*Pasetto et al., 2018*), and even coupled with passive acoustic monitoring (*Elise et al., 2022*), none to our knowledge has yet evaluated the applicability of these datasets for answering local community ecology monitoring questions. We recommend further comparison of local weather station data with satellite remote sensing data, in the context of local biological monitoring programs for evaluating the extent of applicability of this approach.

## CONCLUSIONS

Astronomical moonlight and satellite remote sensing precipitation data can explain the acoustic activity of katydids and crickets respectively in an orthopteran assemblage, and its use may be beneficial for studies with geographical, financial, or other constraints. Still, we consider that further analysis including multiple sampling points is necessary before

generalizing the patterns observed here, given the small time and space scale of the present study. The effects of small changes in environmental factors on species acoustic activity observed in our study suggest orthopterans could be a successful key indicator of environmental change. How those changes could be extrapolated to annual seasonality and other trophic levels is a potential endeavor to better understand acoustic communities worldwide. Finally, acoustic monitoring of orthopterans has a high potential for environmental assessments, in addition to answering ecological questions and enriching taxonomic descriptions of under-studied biotas.

## ACKNOWLEDGEMENTS

To the late Germán Amat García for his encouragement and support as our professor, we hope we made you proud. We acknowledge Beatriz Salgado, Paula Caycedo, Oscar Cadena-Castañeda, and Jose Luis Benavides-López for contributing with ideas and perspectives in early stages of this project and ongoing collaborations. We thank Rainforest Connection-RFCx for allowing us to use the ARBIMON platform for academic purposes without payment. We received the support and encouragement of working in Los Tucanes Natural Reserve by Fernando and Pablo Forero, and their families, as well as Zuania Colón-Piñeiro during her time coordinating the research activities in the reserve.

In addition, we thank the many constructive comments on the manuscript by Zuania Colón-Piñeiro, Sarah McGrath-Blaser, Laurel Symes, and other two anonymous reviewers. We also acknowledge David Gray for providing additional comments, and for encouraging this submission. Precipitation data used in this paper was downloaded from the Giovanni online data system, developed and maintained by the NASA GES DISC.

### Funding

The authors received no funding for this work.

### Competing Interests

The authors declare that they have no competing interests.

### Author Contributions

- Diego A. Gomez-Morales conceived and designed the experiments, performed the experiments, analyzed the data, prepared figures and/or tables, authored or reviewed drafts of the article, and approved the final draft.
- Orlando Acevedo-Charry conceived and designed the experiments, performed the experiments, analyzed the data, prepared figures and/or tables, authored or reviewed drafts of the article, and approved the final draft.

### Field Study Permissions

The following information was supplied relating to field study approvals (*i.e.*, approving body and any reference numbers):

Reserva Natural Los Tucanes.

## Data Availability

The measurements and results and complete monitoring recording dataset list with download access are available in the Supplemental Files.

The original directional recordings are deposited in the Instituto Humboldt's sound collection-Colección de Sonidos Ambientales *Mauricio Álvarez-Rebolledo* (IAvH-CSA-18783 to IAvH-CSA-18805).

## Supplemental Information

A Spanish language translation of this article, provided by the authors is available here. This translation was not reviewed by PeerJ staff or the Board and is the authors' own rendering of the article.

Supplemental information for this article can be found online at http://dx.doi.org/10.7717/peerj.13969#supplemental-information.

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
