# Peer review of "Satellite remote sensing of environmental variables can predict acoustic activity of an orthopteran assemblage"

_PeerJ, doi:10.7717/peerj.13969_

## Round 0.1 · original submission · Major Revisions

Dear Dr. Gomez-Morales and Acevedo-Charry:

Thanks for submitting your manuscript to PeerJ. I have now received three independent reviews of your work, and as you will see, the reviewers raised some concerns about the research (and manuscript). Despite this, these reviewers are optimistic about your work and the potential impact it will have on research studying orthopteran biology and ecology. Thus, I encourage you to revise your manuscript, accordingly, taking into account all of the concerns raised by all three reviewers.

In general, the reviewers wish to see improvements to English and grammar, as well as a better presentation of your findings, particularly regarding clarity.

The methods should be clear, concise and repeatable. Please ensure this, and make sure all relevant information and references are provided.

There are many minor suggestions to improve the manuscript. Note that reviewer 3 kindly provided a marked-up version of your manuscript.

Good luck with your revision,

-joe


Reviewer 1 ·

Basic reporting

A very clear paper, with excellent structure and detail.

Some minor comments on text:

- Line 82: "...in their wings, namely stridulation." -- Stridulation is not always involving wings (see Baker and Chesmore 2020). Authors should check if they wish to rephrase this slightly.

- Line 159: "pattern matching" - the term "pattern matching" is the term used in the software ARBIMON, but it is a little vague. I think this is the same method often referred to as "spectrogram template matching" - authors should consider clarifying what method is used here. (It is possible to use either the waveform or the spectrogram to match a template, for example.)

- Line 373 "to lowering that risk" -> "to lower that risk"
- Line 384 "only that in a different fashion" -> "only in a different fashion"
- Line 413 "repertory" -> "repertoire"
- Line 417 "moonlight make not any difference" -> "moonlight would not make any difference"

Experimental design

This exciting combination of remote-sensing environmental variables, with passive acoustic monitoring, is new to me, a very interesting approach which I think may be new to the field. As the authors indeed mention, the microhabitat conditions may be different from the remote-sensed conditions, but that does not defeat the study (it mainly means that there may be additional environmental correlations as well as the ones they have detected).

The bioacoustic methodology, including acoustic recording and preparation, identification of sonotypes, confirmation of species identity, has a lot of clear detail which gives a lot of confidence in the results.

However, I have one important concern about the acoustic analysis, which I think must be addressed:
- Line 217 and Figure 2 "maximum frequency": how is "maximum frequency" measured? It is clear from Figure 2 that species "Gr8" has a lot of energy in the spectrogram above 10 kHz, yet the measured maximum frequency is less than 10 kHz. (The same is true for other species.) It appears maximum frequency is measured using software "ocenaudio" with which I am not familiar, but it is a general-purpose audio editor. Its measure of "frequency" may thus be designed for human voice/music, and not an appropriate measure of maximum frequency for insect sounds. I suspect that the measure is something like "maximum of the fundamental frequency", where the fundamental ("F0") is quite standard in human voice but nontrivial for insect sound.
The authors must clarify why the "maximum frequency" does not seem to relate to the upper limit of the bandwidth for their species, and whether it is indeed a meaningful measurement for these orthopteran species. The authors may find that they need to use a different software algorithm to measure "maximum frequency" and other frequency statistics.

One minor issue:
- Line 313--318: precipitation affecting singing behaviour. It is commonly observed that precipitation can affect the quality of automatic detection. Can the authors specify how they can be confident that the observed correlation is truly about a change in vocal activity, and not a change in detectability?

Validity of the findings

The paper and supplementary data are admirably complete, and give strong confidence in the results. The discussion is a very good synthesis of the findings.

Reviewer 2 ·

Basic reporting

Overall, I thought this was a very nice study. The main purpose was clearly stated, and the study demonstrates a unique role for remote sensing in studying responses of acoustic animals to environmental conditions. The introduction and discussion do a good job of summarizing and citing the relevant literature on this topic. For the most part, the figures were simple and clear, and the raw data were provided. However, I have concerns about some of the data, either because the methods do not clearly explain how the data were collected, do not give sample sizes, or the sample sizes were too small for the conclusions that are drawn from the data.

There are parts of the manuscript where the writing or organization could be improved, and I have given specific examples of this below. These are mostly simple to fix because the meaning of the sentence is clear, but in a few places I was not sure what was intended, and the authors will need to clarify.

Title: This is nitpicky, but the remote sensing does not explain the acoustic activity. I think a more appropriate wording would be “Satellite remote sensing of environmental variables can be used to predict acoustic activity of a tropical orthoptera community.”
Line 22: The first sentence needs to be restructured. The first half of the sentence ends by saying it is a promising method, but it does not state what the method is for (only implied by first half of sentence). It also states that it is less intrusive, but it does not give the alternative to which is it being compared (i.e. less intrusive than collecting specimens?).
Line 26: What do you mean by “far behind”? Not as advanced as these other studies in terms of methodology? Or just fewer studies?
Line 40: Change “simultaneously” to “same time of night.”
Line 44: I think “shifts” should be replaced with “differences”
Line 47: This makes it sound like it is more likely to rain at dusk than during the night. If you meant something else, the sentence should be reworded.
Lines 60-62: It’s not clear to me what the difference is between these three recording methods. In all cases, aren’t people just recording sounds in the environment? Explain how these studies differ from each other in methods more clearly. For example, what is the difference between acoustic monitoring and direct in situ recordings? The term “in situ” could mean different things unless it is being directly compared to a situation that is not in situ (i.e. lab vs. field). I would remove this term throughout the manuscript and replace it with more specific wording.
Line 65: “It consists of…” should be the start of a new sentence
Line 67: “Arranges” does not make sense here. Did you mean “methods”?
Line 69-73: Split this into two sentences to make the meaning easier to understand.
Line 75: Replace “in situ” with “from the same location.”
Line 77: Replace “in varying” with “that varies in”
Line 79: Again, using a comparative word like “higher” requires an explanation of what is being compared. Insects are more sensitive than what? Mammals / birds? How is this measured or compared? It would be enough to say that they are sensitive to environmental change.
Line 83: Replace “namely stridulation” with “a behavior called stridulation.”
Line 109: “Community temporal patterns” is too vague. Does this mean circadian or long-term patterns or both?
Line 116: I think “spare” should be “sparse”, meaning low density.
Line 118: Change to: “… in such cases because it is spatially and temporally comprehensive…”
Line 130-132: This sentence is confusing. It’s not clear what “went beyond” is referring to, and the second half of the sentence refers to remote sensing, which was already described in the previous sentences. Maybe just replace with “In addition, we found and recorded insects in the same location to identify sonotypes to species (when possible).”
Line 145: I do not understand what is meant by “forest few meters away from the edge.”
Line 151: What is n? Number of recordings? Add this to the sentence.
Line 174: The last part of this sentence should be in brackets
Line 180: Should “temporal” be “temporary”?
Line 182: The codes used for species (e.g. Gr8) in the paper are not very reader-friendly. I understand that the number represents the code you used when classifying the sounds, but it is very difficult for a reader to remember which random numbers are associated with each species description. If you want to use abbreviations for the species, I would recommend making this something more intuitive like Cr1 and Cr2 for the two crickets species and Ka1-5 for the katydids.
Lines 167-168: Many readers will not understand what “overlap of kernel densities” means. More information is provided in a later section of the methods (lines 208-215), but this term needs to be explained to readers when it first appears in the manuscript.
Line 217: Remove “For the…” from the start of this sentence and put a comma after “dominant frequencies” to make this sentence grammatically correct.
Line 232: What are Detections(a)?
Line 274: Replace “still” with “although”
Line 300: What do you mean that the ultrasonic harmonics were not considered? They could not be recorded because of the sampling rate. Also, don’t you mean that they are low amplitude, and therefore can be excluded from consideration?
Line 308: What is the word “other” referring to here?
Line 313, 316, 420: “Detrimental” means something that is harmful. Just say that there is a negative relationship.
Line 317: “… daily precipitation is a better explanatory variable for the …”
Line 322: Replace “drives” with “driving”
Line 326: “… each sonotype” i.e. singular.
Line 334: Light trapping can also work for some species.
Line 342: “Diurnal species…” should be the start of a new sentence.
Line 359: Replace “cluster” with “disperse”
Lines 365-368: There can be overlap in size between crickets and katydids, and the size and properties of the sound producing structures are related to frequency in crickets (and probably katydids) rather than the body size: Godthi, V., Balakrishnan, R., & Pratap, R. (2022). The mechanics of acoustic signal evolution in field crickets. Journal of Experimental Biology, 225(Suppl_1), jeb243374.
Line 381: This is the first time the term “sunset chorus” is used.
Line 384: Remove “…, only that in a different fashion;”
Line 412: Fig. 2 in Romer et al. shows that the number of acoustic signals produced by the focal species does not change with moonlight. The main difference is that these katydids produce more tremulations during full moon. They did find that background noise was less during full moon, which would be a combination of all nocturnal acoustically signaling animals. I think your data is stronger support that katydids are calling less with more moonlight because you have quantified individual sonotypes.
Lines 425-430: Good point.
Fig. 2: Label the species on the spectrogram so that it is clear where one species’ call ends and the next begins.

Experimental design

I thought the combination of PAM and collecting local insects to find out who was producing the sonotypes was a great approach for understanding a community in which the species and songs are not yet well-described. I also appreciated the use of remote sensing data to assess how environmental variables might influence calling behavior in these insects. I think this will be a great example for future studies on this topic.

There were several places, however, where I think the methods require more information or more data are needed.

1. Why was Random Forest used for automatically identifying some species and Pattern Matching for others (lines 158-161, Table S1)? I could not find an explanation for this in the methods. How might this influence the results? When first reading the manuscript, I thought there was going to be a comparison of the two approaches for each species.

2. I don’t feel like recording a single individual and measuring acoustic properties for 5 syllables of song is sufficient for generalizing about calling song parameters of a species (Lines 219-223, SD10). It is true that calling song is very stereotyped and species-specific in insects, but individuals can vary and a sample size of one individual per species leaves open the possibility that it was not a very representative individual for the species. If these measurements were only taken to identify the source of the sonotypes in the field recordings, it might be ok, but it seems like these few values are then used to make statements about general patterns across species (lines 36-39 of abstract, large parts of the results, Fig. 2 & 3). I know that it is very difficult to find these insects in the forest. Were you able to find and record more than one individual per species? If so, I would take more measurements. If not, you could also take acoustic measurements for each sonotype from the field recordings. Audiomoth microphones are not going to have a great frequency response, so that should be acknowledged, but if the acoustic properties are similar to what you recorded from focal individuals, then you would have a much larger sample size for describing acoustic properties.
Also, if I understand correctly, these 5 measurements were randomly sampled 10,000 times to get the mean and confidence intervals for each species? Why was this approach used instead of just calculating the mean and SD? I don’t think that this approach compensates for there only being 5 measurements.
How was minimum and maximum frequency measured? Lines 217-220 say it is taken using FFT analysis and later it says at -10dB. Does this mean that the minimum and maximum were determined as being -10dB below the frequency with the most energy in a power spectrum? How did you deal with the harmonics?

3. The results states the approximate height that each species calls from in the forest (lines 256-270), but the methods do not explain how this was measured. For example, how did you know that the Cocconotini sp. did not sing higher than 4 m? Likewise, no sample sizes are provided for these values, leaving the reader to wonder how well supported these generalizations about singing locations are. I did enjoy reading the natural history observations, like the crickets calling from rolled leaves and the carved burrows of the katydid species.

Validity of the findings

The main strength of this study is the combination of remote sensing data to assess how environmental variables contribute to daily patterns of singing activity in orthopteran insects. The data in Table 2 and Figures 1 and 4 appear to be appropriate measurements of daily activity and environmental variables, and even though they are not collected on fine spatial or temporal scales, they still show very interesting patterns and differences between the species that will be valuable contributions to the literature.

The acoustic analysis of the species, however, is not sufficient for the conclusions about acoustic niche partitioning in this community. Too few acoustic measurements were made on the calls, and the methods and sample sizes were not provided for the microhabitat measurements (see comments above). Therefore, the discussion about species that use the same frequencies at the same time being separated in space (lines 396-400) is not sufficiently supported with quantitative data.

Likewise, comparing the amount of time singing to the frequency of the calling song between crickets and katydids with no phylogenetic comparative methods is not sufficient to suggest that there is a relationship between these two variables. Crickets are known to have a very narrow range of frequencies at which a female will recognize the song as coming from a male (Kostarakos et al. 2009) and are limited to producing low frequencies because of the “clockwork” mechanism of their two resonant wings (Jonsson et al. 2021), which is not true for katydids, so there are morphological and sensory limitations that can explain these differences. When looking at just the katydid species in Fig. 3, there does not seem to be a correlation between frequency and time calling, so this relationship is being driven by two groups of insects that have very different limitations on the frequency of their song. The Symes et. al. (2021) study found a tradeoff in time spent singing and the amount of sound produced per call. Crickets have very high duty cycle song compared to most Neotropical katydids, so the energy put into singing could also be a reason for the difference in circadian rhythms for these species.

In addition, two of the species could not be captured because they live in the canopy, so it seems like the frequency of their song was being used to identify them as either a cricket or a katydid, making the interpretation a bit circular. I think all of these points are very interesting and could be brought up in the discussion, but without more data and analysis, I don't think that they should not be presented as main findings of the study.

Kostarakos, K., Hennig, M. R., & Römer, H. (2009). Two matched filters and the evolution of mating signals in four species of cricket. Frontiers in Zoology, 6(1), 1-12.
Jonsson, T., Montealegre-Z, F., Soulsbury, C. D., & Robert, D. (2021). Tenors not sopranos: bio-mechanical constraints on calling song frequencies in the mediterranean field-cricket. Frontiers in Ecology and Evolution, 9, 225.

·

Basic reporting

The authors integrate remote sensing information and acoustic recording and analysis to generate new insights about insect signaling in a site that has rarely been studied.

This paper is an interesting addition to the literature and a good fit with the selected journal. I have provided line by line comments and grammatical suggestions in the attached document. Apologies for the hand-written responses in the document, it was difficult to insert comments in the PDF and I wanted to return the manuscript without additional delay.

>Line 246: "We identified seven orthopteran species for the acoustic community. Two species of crickets
(superfamily Grylloidea): “Flutist” Gr2 and Podoscirtinae (Gr4) and four katydids (family
Tettigoniidae): Copiphora colombiae (Gr8), Neoconocephalus brachypterus (Gr13), Cocconotini
(gen. nov.) (Gr22), “Sprinkler” (Gr12), and “Rattler” (Gr20)."
I believe that this corresponds to five species of katydid, is that correct?

Experimental design

In the automated detection framework, the authors describe precision metrics (how often the detection was the correct species). Is it possible to provide any estimate of recall (how often various species were missed by the detector)? Missed detections are particularly likely to occur when calls overlap, which is relevant to the hypothesis about acoustic partitioning.

I enjoyed and appreciated the emphasis on acoustic active windows. This is an interesting and informative approach.

>The abstract mentions Trend iii) “calling activity increases proportionately with dominant frequency”
This is a very cool finding! The way that it is phrased in the abstract could be interpreted several different ways. I would say specifically that ‘acoustic activity span’ increases proportionately (otherwise ‘calling activity’ could mean number of individuals or number of calls). Do you have enough data to test it statistically rather than having to say that it “seems to increase”?

>Hamming window 1024 bins
Is this 1024 bins or 1024 samples?

The AudioMoths captured one minute of sound per half hour. For Gr8, the period of high activity is described as 20:00-20:30, but it could equally well be 19:30-20:00. Perhaps it is clearer to report the time stamp of minutes with high activity rather than the 30 minute block.

I found figure 4 quite informative! In Fig 2, I would recommend changing the FFT size to highlight pulse structure (perhaps 256 or 512 samples rather than 1024).

Validity of the findings

The authors of the study focus on the signaling activity of seven orthopteran species. The authors are able to capture several of these species and match them to their sounds. Starting with common species is a valuable way of beginning to understand the behavior and identity of the species in community. However, I think that it is important to consider that these species are a non-random subsample of the community and to address this fact more thoroughly in the discussion. In particular, the paper focuses on the behavior of species that produce many calls (and calling rates range widely across katydid species). The paper also focuses on species that can be recorded using a sampling rate of 48 kHz. Many Neotropical katydid species have calls that fall above the 24 kHz Nyquist frequency and would be invisible to the analysis conducted here. I think that it is totally reasonable to use the strategy that the authors used here (of focusing on species with high acoustic activity), but it is important to interpret the results accordingly.

Specifically, you might consider Line 341 (“our focused species were representative”) and whether they are truly representative, or focused on low frequency species that produce a lot of sound (which is also okay as long as it is clear).


I'm not sure how many species of crickets are present vs. the two species that are sampled, but in the abstract I would consider whether it is useful to qualify the result a bit so that readers generalize appropriately. For example "both of the sampled cricket species call at lower frequency for shorter periods of time" as opposed to "crickets call at lower frequency for shorter periods of time."

When you describe species that have elevated activity at at a particular time, what was the criterion for considering that detections were elevated at that time? (e.g. 150% of the mean number of night-time detections).

>Line 352
I don’t follow the logic of how these findings make crickets or katydids more suitable than grasshoppers

Line 408: What is a unit increase in moon fraction light?

Line 420, what is one unit of rainfall?

Additional comments

Optional suggestion: The Gr abbreviation makes it hard for an unfamiliar reader to differentiate crickets and katydids. You might consider adding a prefix or suffix (even a C/K) to help differentiate them for readers. Similarly, in cases where you have matched a sonotype to a known species, I would lean toward using the known latin name in the text rather than the numeric abbreviation. This allows readers that know the species and genera to contextualize the new information rather than referring back to the numeric abbreviations.

---

## Round 0.2 · accepted · Accept

Dear Dr. Gomez-Morales and Acevedo-Charry:

Thanks for revising your manuscript based on the concerns that were raised. I now believe that your manuscript is suitable for publication. Congratulations! I look forward to seeing this work in print, and I anticipate it being an important resource for groups studying orthopteran biology and ecology. Thanks again for choosing PeerJ to publish such important work.

Best,

-joe

Reviewer 1 ·

Basic reporting

Lots of improvements to the text have been carried out.

Experimental design

The bandwidth measurement is now much clearer: well-described and reproducible. I still find it a little concerning that the frequency range measurements don't seem to cover the true bandwidth. The authors note that they deliberately ignore the non-dominant harmonics because they are hard to measure reliably. This is clear method. But I wonder if some readers may misunderstand and infer that "bandwidth" is indeed the bandwidth of the entire signal. (It also affects the interpretation of the interspecies spectral overlap calculations. But the authors' discussion section does not make any false claims about lack of overlap.)

Overall, I am content to accept this, given that the measurement method is now clearly documented.

Validity of the findings

Good.

Additional comments

I am very pleased with the improvements made in this version of the paper, and I look forward to seeing it published.